# OpenReview forum: "Cognitive models can reveal interpretable value trade-offs in language models"
_ICLR.cc/2026/Conference — ICLR 2026 Poster_

### Official Review · Reviewer_tHgM · 2025-10-21

**Soundness:** 2
**Presentation:** 2
**Contribution:** 3
**Rating:** 4
**Confidence:** 4

**Summary:**

This work introduces a cognitive model of polite speech as an interpretability tool to quantify value trade-offs in LLMs, formalizing the balance between informational, social, and presentational utilities. The authors apply this method to analyze both frontier black-box models under different reasoning budgets and open-source models during RL alignment. The results demonstrate that models' utility preferences shift predictably in response to goal-based prompts, revealing a signature for sycophancy, and show that base model choice has a more persistent impact on these trade-offs than the specific alignment method, with utility values converging early in training.

**Strengths:**

The research method in this paper is very interesting, by estimating the parameters of cognitive models, it enables interpretable analyses of the internal value trade-offs within LLMs.

The paper also compares the analysis results of human cognitive models, providing valuable reference points.

**Weaknesses:**

As the authors discussed in the conclusion, I have some concerns about the methodology of using human cognitive models to analyze LLMs’ value trade-offs. This approach can at best offer correlational explanations, but it essentially does not move beyond treating LLMs as black-box models. If the proposed cognitive model could be mapped to some kind of circuit within the LLM, I believe it would be more convincing.

There are numerous real-world scenarios involving value trade-offs. While I agree that politeness is indeed an important one, focusing only on this type of scenario feels somewhat limited.

The experimental analysis is rather rough. First, the figures are not clear enough to allow readers to intuitively understand the results. Second, there is a lack of deeper analysis, the paper seems to merely present evaluation results without offering further insight.

There are also some minor typos. For instance, a period should follow paragraph names, and punctuation should be added after equations. The use of quotation marks in LaTeX is incorrect (e.g., Line 262). In addition, the $\phi$ symbol in the footnote on Line 377 did not render properly.

**Questions:**

See the Weaknesses section.

---

> ### Author Response · Authors · 2025-11-21
>
> Dear Reviewer tHgM,
>
> Thank you for your time in engaging with our work. We are glad to hear that you found our research method very interesting and recognize its ability to enable interpretable analyses of the internal value trade-offs of LLMs against valuable human reference points.
>
> We hope you’ll consider the following responses to your other concerns:
>
> To address your first point about our methodology treating the LLM as a black-box: we completely agree that mapping the cognitive model to representation space would be really interesting. However, we believe that such analyses would, first and foremost, warrant a separate paper, but also that our current work is a necessary precursor to it. The importance of first validating the existence of the behavior you want to probe for is well-accepted in the mechanistic interpretability literature (we like Krakauer et al., 2017’s work motivating the same for neuroscience), and our methodology offers an avenue for doing so in a consistent, interpretable way so that we don’t spend time probing epiphenomena in models. We see our method as complementary to representational-level analyses by providing a tool for more principled, theory-driven mechanistic interpretability and are familiar with recent works in this area (Prakash et al., 2025, Geiger et al., 2024, and Shai et al., 2024 to name a few).  Informed by these proposals, we have a number of thoughts on how to connect cognitive models to LLMs’ representation space, which we look forward to the opportunity to include in a revised discussion section.
>
> Additionally, while we agree that mechanistic work is important, we believe that this does not take away from the importance of rigorous methods for black-box LLM evaluations. As you highlight, our methodology provides a formal and interpretable alternative to anecdotal or “vibes” based accounts of important behaviors in the closed-source LLMs that most people are currently interacting with. Given that representational-level analyses are not practical in these models, we believe that our work makes an important contribution to tools for understanding and diagnosing their behaviors. While our results may not show casualty at the representation-level, the fact that our prompt- and training-based manipulations all surface intuitive, interpretable parameter values inspires our confidence that our present results are more than just correlational at an independently valuable level of analysis (behavior).
>
> To your second point about the limitations of politeness as a domain for the many real-world scenarios involving value trade-offs, we agree that the work would benefit from expanding the settings and behaviors studied, which we are excited to do given our strong findings about the viability and versatility of this research method, across a number of relevant features of different model settings in an alignment-relevant domain. We look forward to having the extra space of the revision to elaborate on our ideas for expanding this method to real world settings such as delivering bad news, declining a request, or expressing disagreement in sensitive discussions. Our cognitive model methodology is definitely amenable to this in principle and would be taken care of under the same proposals for future work that we highlight in lines 452-456 of the discussion. Our initial attempts to do so revealed technical challenges at a few points that we chose to defer to future work, as we felt that addressing these technical points of cognitive modeling within this paper would distract from our focus on interpreting value trade-offs as a result of LLM design decisions.
>
> Finally, with regards to your concerns about the experimental analysis being rough and not offering further insights, we hope you will consider our above comments about the value of rigorous black-box evaluation methods in their own right. We greatly appreciate you noting that our research methodology enables interpretable analyses of the internal value trade-offs within LLMs and would like to emphasize this our primary goal, which we took care to enact via careful prompt- and training- manipulations that corroborate many existing findings. Additionally, while we put significant effort into the figures, we would welcome any specific suggestions you have for improving them for the clarity of our paper and will endeavor to implement them in the updated version. In the absence of further details, we’re not sure whether this comment refers to the figures themselves, their organization in the paper, or something else. We notice that the other reviewers did not have any issues with the figures themselves, though one reviewer did point out that the placement of Figure 1 was inconvenient to parse, and have already prepared a more intuitive progression of figures to include for the final version. Thank you for pointing out those typos as well, we will work on fixing them in the updated version!

---

> > ### Comment · Reviewer_tHgM · 2025-11-24
> >
> > Thank you for the authors’ response. My biggest concern remains that the polite scenario is too limited. Although the authors have explained the difficulty of extending the method to a broader range of scenarios, I still believe that relying solely on the polite setting is insufficient to meet the quality standards of a strong manuscript.
> >
> > In addition, although the authors have clarified the rigor of their method as an approach for evaluating and interpreting black-box models (and I acknowledge this rigor), I still think that correlation analysis based on such a small number of parameters is inadequate for explaining complex systems like LLMs.
> >
> > Regarding the presentation of the experimental results, I hope the authors can provide a deeper analysis rather than merely presenting evaluation results without offering further insight.
> >
> > On the issue of figures, I share the view of another reviewer that the placement of Figure 1 is inappropriate, and I look forward to the authors correcting this in the revised version.

---

### Official Review · Reviewer_9j7r · 2025-10-31

**Soundness:** 4
**Presentation:** 3
**Contribution:** 3
**Rating:** 8
**Confidence:** 3

**Summary:**

In this work, the authors use the cognitive model of polite speech, a newer version of the influential rational speech act (RSA), to understand value tradeoff in LLMs. The RSA model provides a normative framework to model the weights an agent, human or LLM, attributes for different utility functions in speech, such as information utility, social utility, presentational utility. The authors began by conducting a vignette study on  LLMs -- drawing on experiments run on humans -- and fit RSA model to the LLM generated responses. They found that by default reasoning LLMs, particularly the closed-source ones, tended to weigh informational utility higher than social utility, with the utility values being amenable to prompting methods that prescribe the goals more explicitly. They additionally analyzed how these utilities change over the course of fine-tuning on two different open-source models and found that utility values shift quite early on in the training process with the effect of base model and pre-training data out weighing those of feedback dataset and alignment method.

**Strengths:**

Although the idea of using cognitive models to study the behavior of LLM models is not new, the setting the authors have chosen to study and the model of choice serves as a great example of how one can use cognitive modeling to draw insights about LLMs.  I find both the setting and model to be ecological for the study of value trade-off in LLMs. The findings are intuitive and corroborate many existing findings.I also liked the fact the they considered both open and closed models, and used them appropriately. For example, by using them in cases where certain class of models can be helpful to draw relevant conclusions. The writing was clear, figures legible, and analyses well motivated.

**Weaknesses:**

The RSA model explanation is concurrent a bit too dense, especially for ML audience. It would be extremely helpful if the authors can dig into the model a bit more, with examples to provide readers with more intuition and also interpretation of what different values of the fitted parameters mean. This is especially important as the rest of the papers build on the fact that the readers understand the model and its parameters well.
The authors could have also motivated the datasets used for fine-tuning a bit better, focusing on how they are useful for the answering the questions of interest.

**Questions:**

How well do these models fit human data?
Are there other competing models that can used to compare against?
What happens in these measures if you have more than one back and forth? Can the cognitive model update its parameter with additional observations? It would interesting to see how these measures change over the course of a short, constrained conversation.

---

> ### Author Response · Authors · 2025-11-21
>
> Dear Reviewer 9j7r,
>
> We were happy to hear that this work was seen as a great example of using cognitive modeling to reason about LLMs, and for the other positive comments – thank you!
>
> We also appreciate the feedback about making the RSA model description more accessible for an ML audience, and we agree that this is an important foundation for the rest of the paper. This comment echoes that of another reviewer, and we will work on improving this by incorporating more examples to build intuition around the different components of the model and putting greater emphasis on the interpretation of the parameter values.
>
> We will also include more intuition behind our choice of feedback datasets for the camera-ready version: in particular, we aimed to have very different characteristics captured by each of the datasets to aid the interpretation of the final results. Since UltraFeedback is a synthetic feedback dataset designed to optimize for instruction following, helpfulness/informativity, truthfulness, and honesty attributes, we hypothesized that this dataset would induce a stronger weighting on informational utility in models than HH-RLHF, which more strongly indexes on the harmlessness attribute that might be more strongly associated with social and presentational utilities.
>
> Please see below for more detailed responses to your questions:
>
> * Fit to human data: Yoon et al. (2020) perform ablations across different versions of the cognitive model, and find that the full model (which we use in our work) provides the best fit to the human data, explaining 97% of variance.
> * In addition to the ablations considered by Yoon et al. (2020), Caracassi & Franke (2023) do propose an alternative model of polite speech that largely builds off Yoon et al. (2020), but reframes the presentational utility as “strategic truth stretching” on part of the speaker. We chose to go with the model we did because we believe that the assumption that current LLMs can potentially represent a presentational goal is more reasonable than the assumption that an LLM has an intention to stretch the truth or lie. There are, however, promising avenues for comparing different hypotheses of LLMs behaviors, both via existing cognitive models, and by developing new ones, and we would be happy to update the discussion section with more thoughts on this.
> * We agree the multi-turn setting would be interesting to study, and our cognitive model methodology is definitely amenable to this in principle. This would be taken care of under the same proposals for future work that we highlight in lines 452-456 of the discussion. Our initial attempts to do so revealed technical challenges at a few points that we chose to defer to future work, as we felt that addressing these technical points of cognitive modeling within this paper would distract from our focus on interpreting value trade-offs as a result of LLM design decisions. That said, we are actively considering techniques from recent contemporaneous work on scaling cognitive models (see Qiu et al., 2025 and Tsvilodub et al., 2025) and would love to expand on this in the updated discussion section.

---

> > ### Comment · Reviewer_9j7r · 2025-11-25
> >
> > Thanks for your response. I am happy with the general paper and keep my score as is.

---

### Official Review · Reviewer_6un5 · 2025-10-31

**Soundness:** 2
**Presentation:** 2
**Contribution:** 3
**Rating:** 8
**Confidence:** 4

**Summary:**

The authors investigated LLM preference balance between informativeness and social focus in response generation using a pre-established cognitive model of politeness in humans. They analyzed the default behavior of multiple black-box models, as well as shifts across various reasoning budgets and prompt-induced goal variations. Finally, they analyzed the change in preference balance of white-box models during fine-tuning on established RLHF datasets.

**Strengths:**

- Very well articulated and formulated discussion of weaknesses (section 6).

- Important early step in LLM behavior analysis influenced by pre-existing cognitive models of behavior.

**Weaknesses:**

- The experiment design as described arguably only probes the LLM's model of how it would expect others to behave in this scenario (i.e. this is a Theory of Mind task). It is unclear whether this directly predicts model overt behavior. Given appropriate analysis, this might be addressed by comparisons between LLM-as-judge, LLM-as-agent, and LLM-as-assistant perspectives, but analysis along this dimension seems to be absent.

- Section 5.2 provides p-values but does not specify the test being used.

- There is no mention of precautions to control for positional or label biases when prompting the models (i.e. via utterance choice orderings randomization/etc). This is likely non-problematic if all experiments used a shared fixed option ordering.

- It's unclear why you chose to discard the assumption of greater cognitive cost for negated expressions (Appendix C.2) as used by Yoon et al. The explicit need for additional tokens and generation steps would suggest that greater cost is certain for LLMs and thus more important to include than in human cognitive modeling where that increased cost is fully assumed. This is only further exacerbated by the possibility that some utterances may require more than one token. My best guess is that the assumption was made here because of the label-based cloze test methodology, but this may not be a safe assumption, as the cost of internally associating labels to utterances may still be affected by utterance complexity.

- Minor: The intuitive meaning of V(s) should be defined alongside the existing description of equation (2), which is where it is first used, albeit indirectly (via in-text definition of U_soc). In a similar vein, it is difficult and inconvenient to parse the consequences of Figure 1 when it is so far separated from its use in text (section 5.2).

**Questions:**

- Are the changes to utility value from alignment training robust to obliterative fine-tuning? Relatedly, can models that are sensitive to obliterative finetuning be trained to value align and task align simultaneously?

- Did you perform any analysis on variation across framings (LLM-as-[judge/agent/assistant]) for inferred parameter behavior shifts? If not, and if the results in figures 1 and 4 are based on fitting to all three framings in unison, could this have compromised the result? e.g. could the models show strong sycophancy when acting as an assitant but weak anti-sycophancy when acting as a judge or agent, but the assistant framing effectively dominates the other two framings.

---

> ### Author Response · Authors · 2025-11-21
>
> Dear Reviewer 6un5,
>
> First of all, thank you very much for your positive remarks regarding our work, and for the opportunity to clarify the points you’ve raised. We look forward to improving the writing of the final version to make these more clear.
>
> To the points raised:
>
> * Variation across framings: We share your interest in the possible behavior shifts due to these different framings, and did perform these experiments (see Section 4.1 “Manipulations” and Appendix B.4). We note that we also fit the model to each framing’s results separately (for the reasons you point out). This analysis did not find significant differences in the parameter values across these prompt manipulations. So, due to space constraints, we only report the results of aggregating across the framing variations. With the additional space of a final version, we will unpack these results in more detail.
> * Significance tests in 5.2: Thank you for pointing this out. In line 327 we used a z-test to compare the means since the sample size is > 30, and in lines 339-347 independent samples t-tests, which we will make more explicit in the final version.
> Control for positional or label biases in prompts: This is a very valid concern, and we were concerned about this too. We did randomize the order options for every query to prevent positional biases (see the following code). We will make sure to note this in a final version.
>
>         scenario = f"Scenario: {row['vignette']}\n"
>
>         # randomize order options
>         utterances, codes = create_utterances()
>         shuffled = list(zip(utterances, codes))
>         random.shuffle(shuffled)
>         utterances_shuffled, codes_shuffled = zip(*shuffled)
>
>         options = '\n'.join([f" {chr(97 + i)}) It {utterance}" for i, utterance in enumerate(utterances_shuffled)])
>
> * Negated expression assumptions: We do indeed make this assumption for the reason you stated re: the label-based cloze test methodology. Since the LLMs were prompted to select from provided utterance options, we felt that it was more appropriate to not impose this notion of “cost” or “effort” of utterance production in the way it is conceived of in the cognitive model (the speaker’s effort in producing an utterance). We agree that the possibilities you mention would be interesting to explore, though perhaps as more fundamental evaluations about LLMs’ “awareness” of their own production costs. For the sake of the present work, we will endeavor to make our reasoning behind this modeling assumption more clear and point out the considerations you have identified.
> * Obliterative fine-tuning: Our current results show that alignment training shifts utility values over RLHF checkpoints, and that these shifts depend on base model and feedback dataset, but we do not yet test robustness to subsequent task-focused fine-tuning. We believe this is an interesting direction for future work to test whether it’s possible to co-train for stable value alignment while achieving high task performance. Some concrete methods to achieve this include a multi-objective or constraint-based training (e.g., an additional loss objective that regularizes against drifting utilities), retaining some alignment data or synthetic proxies of the polite-speech scenarios, or using lightweight adapters for task-specific finetuning so that ‘obliterative’ finetuning is less likely to affect the original model’s value alignment. Importantly, our framework gives a way to concretely measure potential value alignment drift during fine-tuning. We will note these directions in the discussion of the final version
> * Minor: Thank you for these points! We will work on making the RSA model section easier to understand, and also have already prepared a more intuitive progression of figures to include for the final version.

---

### Official Review · Reviewer_ng39 · 2025-11-01

**Soundness:** 3
**Presentation:** 3
**Contribution:** 3
**Rating:** 8
**Confidence:** 3

**Summary:**

- The paper proposes using formal "cognitive models" to interpret how LLMs handle value trade-offs. Specifically, it applies a model of polite speech that formalizes communication as a trade-off between three competing utilities.
- The authors collect behavioral data by giving LLMs "experimental vignettes" where they must choose a polite (or impolite) utterance in a social context (e.g., judging a friend's bad cake) . By fitting the LLMs' responses to the cognitive model, they infer the underlying utility weights.
- In a study of Anthropic, Google, and OpenAI models, the paper finds that models with higher "reasoning budgets" (e.g., using more tokens or an explicit reasoning mode) demonstrate a higher default preference for informational utility over social utility. These utility weights also shift in predictable ways when the models are explicitly prompted with "informative" or "social" goals.
- The choice of the base model and its pretraining data has a more "outsized" and persistent impact on the final utility trade-offs than the specific feedback dataset.

**Strengths:**

- The authors propose create a behavioral "signature" for sycophancy (hypothesized as high presentational utility but low informational and social utility). They find that when models are prompted with a purely "social" goal, their inferred parameters converge to this "sycophantic" signature---this has some practical use, given increasing worry around syncophancy.
- The experiments are quite thorough, using both open- and closed-source models, running a lot of ablations, and putting detailed results in the Appendix.
- The findings around the centrality of the base model and pretraining data re: sycophancy has important implications for how models will be post-trained.

**Weaknesses:**

- Perhaps it's because I lack a proper cogsci background, but the specific framework used by Yoon et al. that was borrowed by the authors wasn't fully clear to me. The overall approach made sense, however.
- The paper's conclusions about general LLM "value trade-offs" are based entirely on polite speech (i.e., judging a friend's cake or poem) . The authors concede that these cognitive models "are often bespoke to the target domain" and "do not easily generalize to the open-ended nature of natural language use".
- The authors admit that their method has technical challenges. They state that fitting such a complex model "could potentially pose a challenge for making robust inferences".

**Questions:**

None.

---

> ### Author Response · Authors · 2025-11-21
>
> Dear Reviewer ng39,
>
> Thank you for your support of our work. We were glad to hear you found the paper to be useful and thorough, and to have important implications.
>
> To the comments, suggestions, and questions:
>
> * We agree with your comment, which echoes that of another reviewer, that the RSA model description could be made more accessible for an ML audience. We will work on improving this section by incorporating more examples to provide readers with intuitions for each of the components and decisions in the model. We will also include more interpretation of what different values of the fitted parameters mean for the camera-ready version.
> * To the points highlighted by the Reviewer that were mentioned in our discussion: given that we highlighted these points, and that the Reviewer comments do not ask us to undertake the major work of addressing them in *this* paper (if we understand correctly), we did not intend to address them in this paper. However, we would like to note that we are also currently actively working on extensions of this work, that build on this work and address the issues highlighted in the discussion section, in particular generalizing to more open-ended language. We look forward to sharing those results in the future.

---

> > ### Comment · Reviewer_ng39 · 2025-11-25
> >
> > Sounds good! I will maintain my positive score of the paper.

---

### Meta-Review · Area_Chair_epcC · 2026-01-06

**Summary:**

The submission titled "Using cognitive models to reveal value trade-offs in language models" imports tools from cognitive science to evaluate the value trade-offs made by language models, focusing on a particular test of a model of 'polite speech'. The submission places this type of research as rigorous evaluation of a black-box system, complementary to ML-based and mechanistic understanding of model behavior. Most reviewers are convinced by the experimental analysis and use of this model in the context of language modeling and find the analysis insightful.

**Reviewer Concerns:**

There are some remaining concerns regarding the appropriateness of re-using these models for language models, but I do think the submission is sufficiently articulate about the limitations of this approach and its complementarity to mechanistic understanding.

Smaller concerns regard the limitation to studying just one cognitive model, and writing in the experimental section, which the revision only partially addresses.

Overall though, this is an interesting and topical submission that explores the decision-making of language models from a new angle that would be worthwhile to present at ICLR.

**Reviewer Scores:**

ng39, 6un5 and 9j7r remain positive. tHgM has responded to confirm that they still hold their concerns which are outlined above.

---

### Decision · Program_Chairs · 2026-01-26

Accept (Poster)